# Tempo and drivers of plant diversification in the European mountain system

Jan Smyčka [1,2,3✉], Cristina Roquet [1,4], Martí Boleda[1], Adriana Alberti[5,6], Frédéric Boyer [1], Rolland Douzet[7], Christophe Perrier [7], Maxime Rome[7], Jean-Gabriel Valay[7], France Denoeud [5], Kristýna Šemberová [3,8], Niklaus E. Zimmermann [9], Wilfried Thuiller[1], Patrick Wincker [5], Inger G. Alsos [10], Eric Coissac[1], the PhyloAlps consortium* & Sébastien Lavergne[1]

There is still limited consensus on the evolutionary history of species-rich temperate alpine floras due to a lack of comparable and high-quality phylogenetic data covering multiple plant lineages. Here we reconstructed when and how European alpine plant lineages diversified, i.e., the tempo and drivers of speciation events. We performed full-plastome phylogenomics and used multi-clade comparative models applied to six representative angiosperm lineages that have diversified in European mountains (212 sampled species, 251 ingroup species total). Diversification rates remained surprisingly steady for most clades, even during the Pleistocene, with speciation events being mostly driven by geographic divergence and bedrock shifts. Interestingly, we inferred asymmetrical historical migration rates from siliceous to calcareous bedrocks, and from higher to lower elevations, likely due to repeated shrinkage and expansion of high elevation habitats during the Pleistocene. This may have buffered climate-related extinctions, but prevented speciation along elevation gradients as often documented for tropical alpine floras.

[1] Univ. Grenoble Alpes, Univ. Savoie Mont Blanc, CNRS, LECA, FR-38000 Grenoble, France. [2] Center for Theoretical Study, Charles University and the Academy of Sciences of the Czech Republic, CZ-11000 Prague, Czech Republic. [3] Department of Botany, Faculty of Science, Charles University, CZ-12801 Prague, Czech Republic. [4] Systematics and Evolution of Vascular Plants (UAB) – Associated Unit to CSIC, Departament de Biologia Animal, Biologia Vegetal i Ecologia, Facultat de Biociènces, Universitat Autònoma de Barcelona, ES-08193 Bellaterra, Spain. [5] Génomique Métabolique, Genoscope, Institut François Jacob, CEA, CNRS, Université Evry, Université Paris-Saclay, FR-91057 Evry, France. [6] Université Paris-Saclay, CEA, CNRS, Institute for Integrative Biology of the Cell (I2BC), FR-91190 Gif-sur-Yvette, France. [7] CNRS, Lautaret, Jardin du Lautaret, Université Grenoble Alpes, FR-38000 Grenoble, France. [8] Czech Academy of Sciences, Institute of Botany, CZ-25243 Průhonice, Czech Republic. [9] Swiss Federal Research Institute WSL, CH-8903 Birmensdorf, Switzerland. [10] UiT – The Arctic University of Norway, The Arctic University Museum of Norway, N-9037 Tromsø, Norway. *A list of authors and their affiliations appears at the end of the paper. ✉email: smyckaj@gmail.com

Mountain regions across the world are important biodiversity hotspots, owing to their high species richness, endemism and faster pace of species diversification compared to lowlands[1–4], with tropical mountain regions harboring by far the highest concentrations of plant diversity[1,5]. The diversity of temperate mountain floras is less prominent at the global scale, but still remarkably high compared to surrounding lowlands, specifically when considering the stressful climatic and edaphic conditions prevailing at high elevations and the dramatic climate and glacier oscillations of the Pleistocene period[6–8]. The exceptional plant diversity and endemism of the European mountains was early recognized, and its emergence has challenged the understanding of botanists and biogeographers since the pioneering work of von Haller 250 years ago[9,10]. Yet, how and when plant lineages have diversified in temperate mountains still remains insufficiently documented. While the drivers (how) and the tempo (when) of species diversification have already been explored for several European mountain plant lineages (e.g.[11–13]), no consensus has yet emerged on the evolutionary history of the species-rich European alpine flora. This is mostly due to previous clade-specific studies using both sparse and distinct genetic data, diverse analytical methods, and as a result, sometimes even yielding non-conclusive results[11–15]. To allow generalizations about the tempo and drivers of alpine plant diversification, we thus need to collect high-quality genomic data covering multiple plant lineages, estimate reliable and comparable phylogenies, and apply a multi-clade framework of modern phylogenetic comparative methods across study lineages.

The tempo of species diversification within mountain biotas is often described as a continuous, typically fast, process following the uplift of mountain ranges[16–18], which slowed down after saturation of the whole available physical and ecological space (e.g.[19]), eventually being modulated by climatic changes. A dominant climatic process that affected current mountain biotas worldwide[11,17,18,20] was the gradual cooling of global temperatures since the middle Miocene (15 Ma BP), that culminated with the onset of Pleistocene (2.6 Ma BP), a geological epoch characterized with low average temperatures and cyclical fluctuation of global temperatures[21]. The effect of Pleistocene could have been positive in terms of net diversification rates with speciation being stimulated by climate-induced range dynamics (speciation pump mechanism[22,23]), or negative with decreased speciation or increased extinction rates due to the impact of Pleistocene glacial periods. The latter perspective is motivated by the observation of greater plant endemism in areas that have experienced relative climatic stability or less glaciation[7,24–26]. Contrary to subtropical and tropical mountains such as the Northern Andes where massive plant diversification occurred during the Pleistocene (e.g.[27,28]), it has long been considered that temperate mountains such as the European Alps were so severely glaciated during Pleistocene cold periods that this posed limits to recent plant diversification[15,29]. However, the evidence for refugia at the peripheries of glaciated European mountain ranges[7,8,30,31] and also on ice-free mountain tops protruding from glaciers (so-called nunataks[32–35]) suggests that unglaciated mountain habitats persisted during glacial periods and may even have triggered plant speciation. Whether Pleistocene climatic oscillations slowed down or spurred the diversification of European mountain plants, and what was the role of pre-Pleistocene gradual cooling, still remains unclear, with most recent reliable analyses concerning Primulaceae only[11]. Documenting the tempo of alpine plant diversification in European mountains therefore requires estimating accurate divergence times for a set of distinct plant clades. This would allow to carefully assess how diversification rates have varied through time and also between regions and environments that have been differently impacted by Pleistocene climate and glacial oscillations.

The unique flora of alpine environments was likely assembled through a complex interaction between spatial and ecological drivers that have influenced the divergence and migration of mountain plant lineages, but the relative influence of these drivers requires further investigation. It has long been considered that allopatric divergence was the main speciation driver in European mountains[11,15,36,37] However, this view is challenged by the evidence that genetic structure and local adaptation across different elevation belts and bedrocks is pervasive in European mountain plants[38,39], and that parapatric speciation along elevational gradient seems to be common in tropical and subtropical mountain floras[1,5,40–44]. In addition, the Pleistocene climatic oscillations may have unevenly impacted the rates of migration, speciation and extinction of plant lineages across the major ecological gradients (e.g. bedrocks, elevation) and between mountain ranges, depending on whether the particular areas were heavily glaciated or remained ice-free during most of the Pleistocene—e.g. high elevation, siliceous areas of central Alps vs. low and mid-elevation, calcareous areas of peripheral Alps, respectively[45]. It thus remains unclear whether species diversification in European mountains was dominantly driven by geographic divergence, or whether it was significantly complemented by speciation events occurring along ecological gradients as known from tropical mountain systems (e.g.[5,41–44]). Moreover, it is not clear how these evolutionary mechanisms interacted with glaciation dynamics, particularly how climatic oscillations influenced diversification and migration in the regions and parts of ecological gradients heavily impacted by glaciation. Disentangling these effects requires simultaneously estimating past rates of migration and cladogenesis within and between mountain ranges, elevation belts and bedrock types, which has never been performed so far on any temperate mountain flora.

In our study, we aim at providing a window into the evolutionary history of European mountain plants by inferring the tempo and drivers of speciation of six study plant lineages considered as representative cases of in situ diversification within European mountains (Fig. 1). These clades are: *Androsace* sect. A*retia* (hereafter *Androsace*), *Campanula* sect. *Heterophylla* (hereafter *Campanula*), *Gentiana* sections *Gentiana*, *Ciminalis* and *Calanthianae* (hereafter *Gentiana*), *Phyteuma*, *Primula* sect. *Auriculata* (hereafter *Primula*) and *Saxifraga* sect. *Saxifraga* (hereafter *Saxifraga*). We reconstruct their phylogenies using a full-plastome phylogenomic dataset obtained by low coverage shotgun sequencing. Using a likelihood-based multi-clade comparative model approach developed specifically for this purpose, we estimate past rates and tempo of species diversification across all study lineages, and also lineage-specific deviations from general trends. We show that overall, there is a weak or no effect of past climate cooling on diversification rates in our study lineages. Further, we use state-dependent diversification models to show that the evolutionary assembly of these plant lineages across different bedrocks and elevational belts was strongly driven by migration. Finally, we examine the importance of different modes of speciation based on sister species comparisons, and demonstrate that the dominant mode of speciation was allopatry, complemented with bedrock-driven speciation. Our results suggest that past migrations across ecological gradients might have buffered climate-related extinctions on one hand, but prevented ecological speciations on the other hand.

## Results

**Tempo of species diversification.** The six study lineages started to diversify in Europe at variable dates within the last 40 Ma (Fig. 2a). The credibility intervals of crown age estimate largely overlap between study lineages, but *Saxifraga* (95% highest

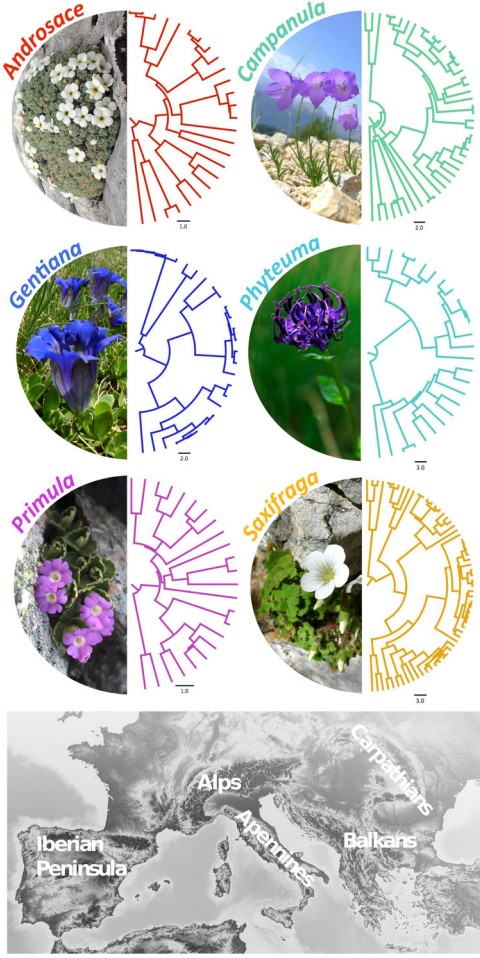

**Fig. 1 Map of the European mountain system depicting the five major geographic regions and the six mountain plant lineages that notably diversified in European mountains.** 84% of all species from these lineages was sampled in this study and we sampled across all five mountain regions. The colors used for the six different phylogenies are used accordingly throughout following figures. The timescale unit of phylogenetic branch lengths is Ma. The background map is adapted from maps-for-free.com.

parameters and compared their combined fit to the multi-clade model. Our multi-clade approach is based on calculating joint likelihood function as a product of individual likelihood functions of models fitted on each of the six lineages (see Methods for details). In order to validate the performance of multi-clade approach and to address general issues with identifiability in diversification models[46], we explored the models behavior with simulated data, showing that our approach can perform unbiased model selection and accurately estimate model parameters (Supplementary Methods 2).

The best performing multi-clade model assumed constant speciation and extinction rates (Table 1). Time- or temperature-dependent models (median AICdiff between −1.16 and −1.71 $_{[df=1]}$ across 100 sets of Bayesian posterior trees) cannot in principle be rejected with confidence, but our sensitivity analysis showed that the used modeling approach and dataset have sufficient statistical power to detect temperature-dependent scenarios where Quaternary speciation rate dropped to 63% of Tertiary rate or less (Supplementary Methods 3). The universal temperature-dependence of speciation across all clades in our dataset considered together was thus either absent or weaker than such drop. Importantly, the combination of lineage-specific models slightly outperformed the models with shared parameters (median AICdiff = 9.68 $_{[df=10]}$), suggesting that the estimated diversification parameters in reality differed across lineages (Fig. 2b). In particular, the diversification dynamics of two lineages was better explained by models with non-constant rates (Table 1): *Primula* showed a slowdown of speciation either with time (median AICdiff = 2.04 $_{[df=1]}$) or during colder periods (median AICdiff = 2.09 $_{[df=1]}$); and *Androsace* showed support for a speciation slowdown in colder periods, although model performance was only slightly better than the constant rate model (median AICdiff = 0.40 $_{[df=1]}$) in this clade.

**Evolutionary assembly: the relative influence of bedrock, elevation and geography.** We used cladogenetic state-dependent diversification models (ClaSSE,[47,48]) to analyze the evolutionary assembly of our study lineages across bedrock types (calcareous vs. siliceous), elevation belts (high elevation vs. mid-elevation habitats), and geographic regions (five major European mountain regions, see Fig. 1). Using AIC comparisons, we quantified the importance of speciation associated with splits between bedrock types, elevation belts, or regions (which we term state-change speciation); speciation within the same bedrock type, elevation belt or region (constant-state speciation); and the importance of change of bedrock type, elevation belt or region occurring along branches of the phylogeny, that is without been linked to speciation events (which we term migration). The ClaSSE models for bedrocks and elevation belts were equivalent to a GeoSSE model[47], while the model for geographic regions represents a generalization of GeoSSE for more than two regions (see Methods and Supplementary Methods 4 for details). The best models inferred from model selection were then re-run in a Bayesian framework to obtain credibility intervals around parameter estimates. As in the previous analyses, all models were evaluated in the multi-clade framework with a model sharing parameters between clades, and subsequently as lineage-specific models with distinct parameter set for each clade. To validate that parameters of multi-clade and lineage-specific models are identifiable, we tested these approaches with simulated phylogenies reflecting the size and structure of our dataset (Supplementary Methods 4).

The state-change speciation between siliceous and calcareous bedrocks appeared to be an important driver of plant diversification (AICdiff = 8.96 $_{[df=1]}$). Model parameters suggest that a bedrock generalist lineage split into descendant lineages specializing on calcareous or siliceous habitats on average cca 0.5 times

probability density of crown age 12.5–38.1 Ma BP), *Phyteuma* (13.7–24.5 Ma BP), *Gentiana* (6.4–28.5 Ma BP) and *Campanula* (10.1–16.5 Ma BP) were probably older than both Primulaceae clades, namely *Androsace* (4.7–10.7 Ma BP) and *Primula* (3.2–7.9 Ma BP). All lineages accumulated considerable amounts of diversity both before and during the Pleistocene (2.6 Ma BP) so that none of them results from exclusively Tertiary nor exclusively Quaternary diversification. Our estimates integrate sources of uncertainty stemming from both molecular and fossil data in a conservative way and seem to be robust to alternative interpretations of fossil record—for details about dating analyses and handling of various sources of uncertainty, see Methods and Supplementary Methods 1.

To better understand the dynamics of clade diversification, we tested a range of hypotheses about diversification changes through time by contrasting five models depicting different diversification scenarios (Table 1). The model support was evaluated by difference in AIC between the focal model and a nested null model not containing the focal parameter(s) (referred to as AICdiff throughout the paper, see Methods for details). We first ran all the models in a multi-clade setup with parameters shared across the six lineages to seek for general patterns. We then separately re-ran the five models with lineage-specific

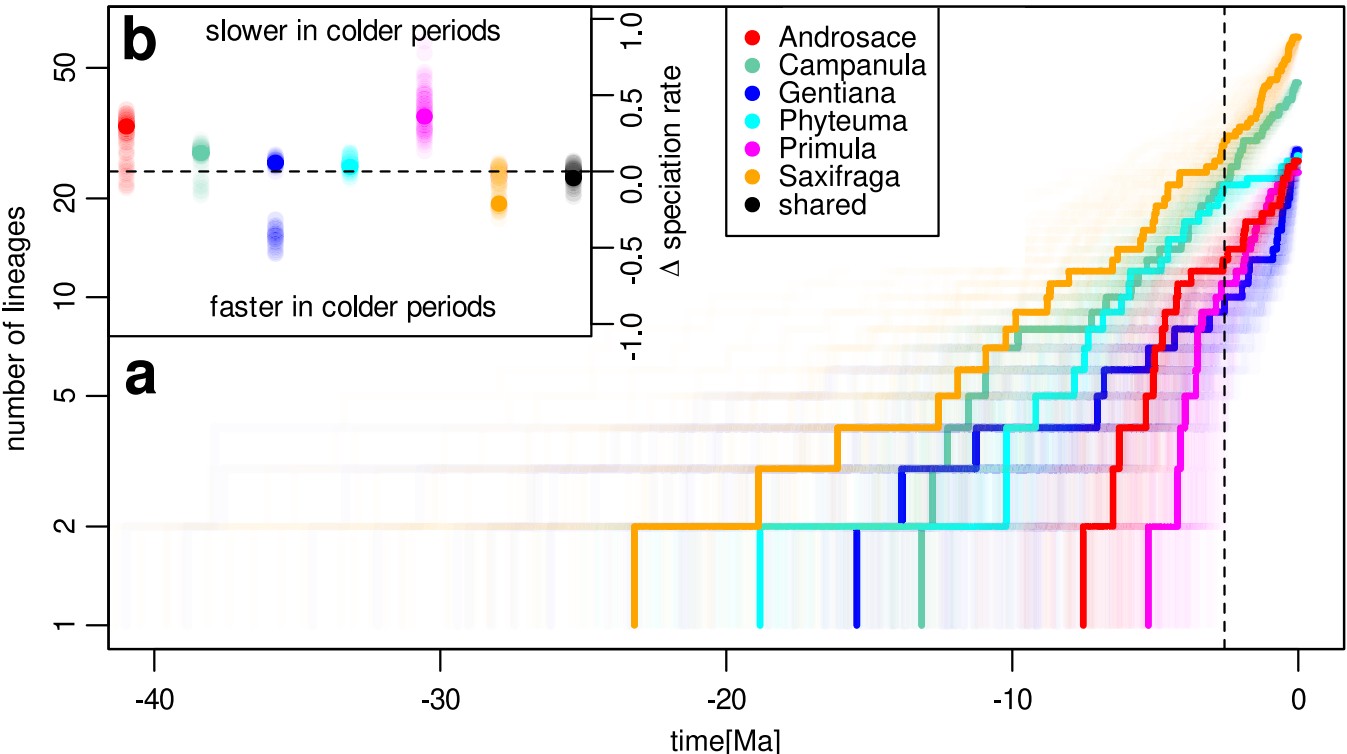

**Fig. 2 Tempo of species diversification for the six study lineages. a** Lineage-through-time curves of each lineage. The thick lines represent maximum credibility phylogenetic reconstructions, while the semi-transparent lines represent 100 trees sampled from Bayesian posterior distributions. The dashed line marks the onset of the Pleistocene at 2.6 Ma before present. The number of lineages is plotted in logarithmic scale, i.e. the exponential growth expected under pure birth model would appear linear here. **b** Parameter estimates indicating the effect size of the temperature control on speciation rates in temperature-dependent models of species diversification (Δ speciation rate). The black dots represent the estimates from the multi-clade model with shared parameters among lineages. Higher values above the dashed line indicate lower speciation in colder geological periods. Thick dots correspond to estimates based on the maximum credibility phylogenetic trees, and semi-transparent dots on the 100 trees sampled from the posterior distribution. Source data are provided as a Source Data file.

per Ma and such speciation events constituted 19% of all speciation events (Table 2). However, we found no evidence for state-change speciation between elevation belts (AICdiff = $-2$ [df=1]) or between the five major mountain regions of Europe (AICdiff = $-1.62$ [df=1]). The latter result suggesting no allopatric speciation between mountain regions is however likely biased due to the coarse scale of the considered mountain regions caused by computational limitations (see Methods for details). For this reason we addressed the prevalence of allopatric speciation and its spatial scaling with complementary analysis of sister species (see below).

Constant-state speciation rates differed between bedrock types (AICdiff = $3.51$ [df=1]) and regions (AICdiff = $17.87$ [df=4]), but not between elevation belts (AICdiff = $-1.02$ [df=1]). In particular, we inferred higher speciation rates on siliceous than on calcareous bedrock (mean estimate and credibility interval in Fig. 3a) and also higher speciation rates in the Alps and the Iberian mountains than in any other European mountain region (Supplementary Fig. 1), but no notable difference in speciation rates between high- and mid- elevation belt (Fig. 3b). Unlike speciation, the inferred extinction rates did not vary between bedrock types, elevation belts or regions (AICdiff < 0 in all the cases). Interestingly, the differences in diversification rates were not the major force affecting proportions of species across bedrock types or elevation belts. In other words, higher speciation rate in certain habitats did not necessarily result in higher species richness in these habitats (Figs. 3e, f). Rather, the difference in contemporary species richness seems to originate from

directional shifts along ecological gradients, as our analyses inferred strong net directional migration between bedrock types and elevation belts. The rate of migration from siliceous to calcareous bedrock habitats was higher than in the opposite direction (AICdiff = $5.59$ [df=1], Fig. 3c), which better explains comparable present-day proportions of species growing on silicate and calcareous bedrocks (Fig. 3e, Supplementary Fig. 2) despite differing speciation rates. We found even more important asymmetric migration rates between elevation belts, as we inferred strong directional migration from high to mid-elevation habitats (AICdiff = $19.79$ [df=1], Fig. 3d). This result captures events of secondary migration of high altitude ancestors toward lower elevations, leading to relatively high present-day and equilibrium proportions of species inhabiting either mid-elevations or occurring in both elevation belts (Fig. 3f, Supplementary Fig. 2).

Models of evolutionary assembly slightly differed among the six study lineages (AICdiff = $7.39$ [df=30] for bedrock, AICdiff = $21.07$ [df=20] for elevation). The most notable outliers concerning the assembly across bedrock types were *Androsace* and *Phyteuma*, which exhibit higher proportions of species occurring on siliceous than on calcareous bedrock (Fig. 3e). In *Androsace*, this was linked to marginally faster diversification rates on siliceous than on calcareous bedrock and symmetrical migration between the two bedrock types. *Phyteuma* showed marginally faster diversification on calcareous bedrock, but strong directional migration towards siliceous bedrock, thus explaining the higher species diversity of this genus on siliceous bedrock

**Table 1 Median AIC differences (AICdiff) between multiple time- or temperature-dependent diversification models and the nested null model with constant speciation and constant extinction.**

|  | Androsace | Campanula | Gentiana | Phyteuma | Primula | Saxifraga | shared |
|---|---|---|---|---|---|---|---|
| Speciation constant, extinction constant | 0.00 | 0.00 | 0.00 | 0.00 | 0.00 | 0.00 | 0.00 |
| Speciation dependent on past temperature, extinction constant | 0.40 | −0.99 | −1.42 | -1.82 | **2.09** | −1.70 | −1.71 |
| Speciation constant, extinction dependent on past temperature | −2.00 | −1.82 | −0.25 | −2.00 | −2.00 | −1.83 | −1.16 |
| Speciation exponentially dependent on time, extinction constant | 0.00 | −0.95 | −0.45 | −1.89 | **2.04** | −1.81 | −1.16 |
| Speciation constant, extinction exponentially dependent on time | −0.08 | −1.53 | −0.31 | −2.00 | 1.62 | −1.80 | −1.25 |

Values with AICdiff > 2 are depicted in bold.

**Table 2 Comparison of ClaSSE models of evolutionary assembly containing parameters of state-change speciation between bedrocks (siliceous vs. calcareous), elevation belts (high elevation vs. mid-elevation) or five coarse geographic regions, with the nested models lacking state-change speciation terms.**

|  | Rate of state-change speciation (Ma$^{-1}$) | Proportion of state-change to all speciation events | AICdiff |
|---|---|---|---|
| **Bedrock types** | **0.453** | **0.19** | **8.96** |
| Elevation belts | <0.001 | <0.001 | −2.00 |
| Geographic regions (coarse scale) | 0.013 | 0.063 | −1.62 |

Values with AICdiff > 2 are depicted in bold.

(Fig. 3a, c). A notable exception concerning assembly across elevation belts was *Androsace*, which has a higher proportion of species occurring at high elevations (Fig. 3f) due to directional migration from mid- to high elevations (Fig. 3d). Based on ancestral state reconstructions (Supplementary Fig. 3b), *Androsace* also contains two exceptional lineages (containing five species each) which most likely spent their evolutionary history exclusively in high elevation habitats. *Gentiana* also has a relatively high proportion of species in high elevation habitats, but here the very broad credibility intervals of model parameters did not allow us to further distinguish scenarios of evolutionary assembly.

**Sister-species overlap in geographical and ecological space.** Due to computational limitations, the ClaSSE models used above could only be run with a maximum of five geographic regions. Hence, we performed a complementary analysis to infer how sister species vary in the dimension of overlap in terms of elevation belt, bedrock type and geographic range, the latter being estimated both with coarse-scale and fine-scale geographic regions (five and 87 mountain regions, see Fig. 1 and Supplementary Methods 5). Most sister pairs diverged over the fine geographic scale (Fig. 4), with 37% sister species pairs showing no geographic overlap at this scale. Similarly, as in ClaSSE models, sister species differed less across bedrock types (27% of sister pairs with no overlap) and over the coarse geographic scale (22% of sister pairs with no overlap). Only 9% of sister species pairs showed no overlap in the occupied elevation belts.

Although the order of speciation modes remained the same among all six study lineages, the degrees of overlap slightly differed between lineages (Fig. 4). Notably, the lineages with lowest geographic overlap at the fine scale were *Primula* (63% of pairs with no overlap) and *Androsace* (43% of pairs with no overlap), suggesting that allopatric speciation at finer geographic scale was particularly important for these two lineages. The lineage *Campanula* exhibited high variance in sister species overlap distribution at fine geographic scale, on the one hand having a high proportion of sister pairs with no overlap (43%), but on the other hand having the highest proportion of sister species pairs showing complete geographic overlap (36%).

## Discussion

Here we explored the tempo and drivers of species diversification across a representative sample of six diverse plant lineages whose evolutionary history is tied to the European mountain system. As we purposely narrowed our study down to clades that previously received a systematic revision, it is clear that many of our studied species had already been sequenced, although always for a few genes only and using variable methodologies. Instead, we employed a uniform next-generation sequencing approach to all study clades and produced over two hundred whole plastid genomes. Chloroplast genomes provide a common ground molecular information that can be applied across various groups and phylogenetic scales. While the resulting topologies may be prone to biases introduced by incomplete lineage sorting or hybridization events in some cases (see Methods for more information and a brief discussion[28]), our chloroplast-based approach avoids some of the pitfalls of nuclear phylogenomics (e.g. paralogy[49]). Our study therefore constitutes a significant step forward toward documenting the evolutionary origins of alpine plant diversity in European mountains, because it is the first to harness next-generation sequencing on multiple plant clades in parallel and to apply a multi-clade comparative framework to explicitly model the tempo and drivers of mountain plant diversification. To make sure that our data collection strategy and analytical approaches can inform us about patterns of mountain plant evolution, we also extensively validated the identifiability and sensitivity of our comparative models using simulated datasets. Our results may thus allow certain generalizations across our study lineages, confirming previous knowledge or botanists intuitions regarding plant diversification in European mountains, documenting some quite surprising, and perhaps counter-intuitive patterns, and finally identifying some interesting variations between the evolutionary histories of different study clades.

The onset of cold and variable Pleistocene climate has caused massive losses of plant diversity in European lowlands[50] and was long thought to have induced a diversification slowdown in European mountains[29], which was also demonstrated with particular plant lineages[11,15]. Surprisingly, we detected neither strong nor homogeneous effects of past temperature on plant diversification rates across our six study lineages. Diversification dynamics in all study clades together were best explained by a

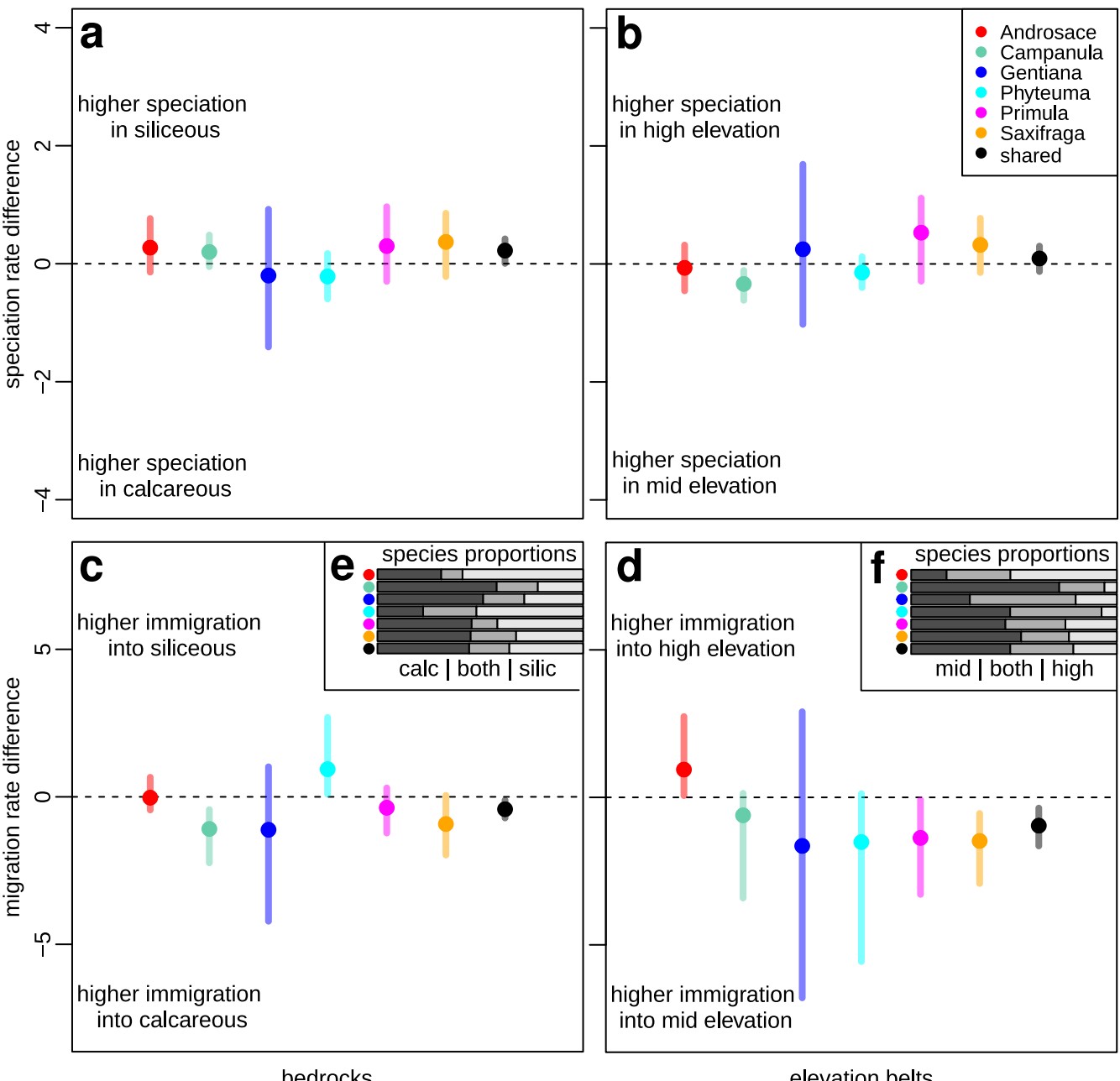

**Fig. 3 Evolutionary assembly across bedrocks and elevational belts based on ClaSSE models. a**, **c** The differences between siliceous and calcareous habitats in constant-state speciation and migration rates, respectively. **b**, **d** The differences between high and mid-elevation habitats in constant-state speciation and migration rates, respectively. The dots represent mean parameter estimates and the bars indicate 95% credibility intervals, based on 5000 MCMC samples from the ClaSSE model posterior. Black dots and bars represent the shared parameter estimates from the multi-clade model, where each of the 6 phylogenies is regarded as an independent realization of the same diversification process. **e** The proportions of species inhabiting siliceous, calcareous or both types of habitats (light, dark and middle gray, respectively), and **f** the proportion of species inhabiting high elevation, mid-elevation or both types of habitats (light, dark and middle gray, respectively). Source data are provided as a Source Data file.

constant rates model, and showed no indication for overall slowdown of species diversification with decreasing global temperature. When analyzed at the level of individual lineages, only the two Primulaceae clades showed certain support for diversification slowdown, as was also demonstrated in earlier studies[11,15]. Yet more interestingly, the other four lineages showed no individual sign of slowdown with decrease of temperature. The past temperature thus did not universally influence the diversification rates of our study lineages, which was not only true for the gradual, and perhaps hardly detectable, effect of cooling since middle Miocene, but also for the onset of

dramatically colder Pleistocene climate. This result was further corroborated by a sensitivity analysis, suggesting that if any temperature-driven Pleistocene reduction of diversification rates took place across all study lineages, it was most likely weaker than a drop to 63% of pre-Pleistocene rates, given the observed data.

The weak and ambiguous effect of Pleistocene climate on diversification rates was also detectable in analyses across elevational gradient and geographical space: the high elevation habitats and also the Alps *sensu stricto*, which were most severely impacted by Pleistocene glaciers, did not show lower rates of

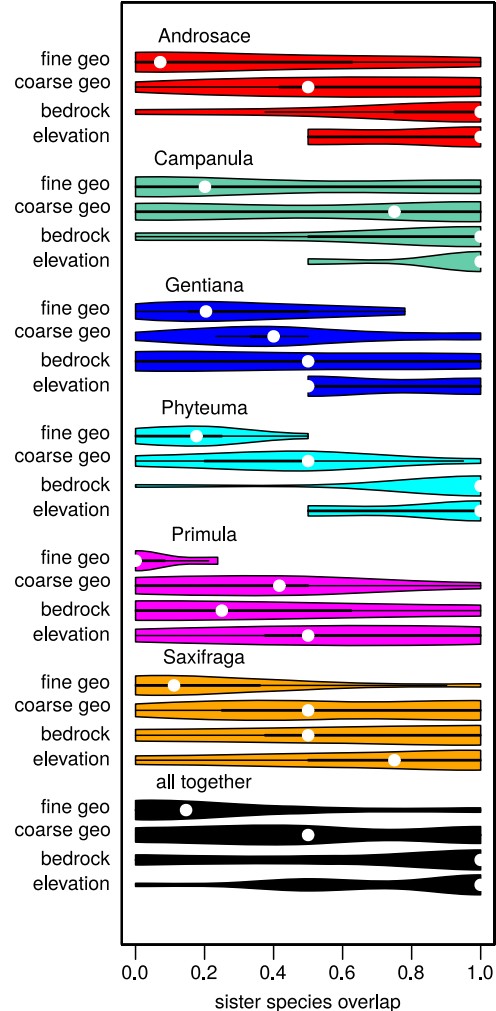

**Fig. 4 Sister-species overlap in geographic and ecological space.** Violin plots show distribution of Schoener's D index of overlap of sister species pairs across 87 operational geographic units (fine geo), five European mountain regions (coarse geo), bedrock types (calcareous vs. siliceous) and elevational belts (high vs. mid-elevations). The point within each plot represents median overlap value. Low overlap between sister pairs is considered as an indication of frequent speciation along the respective geographic or ecological dimension. Source data are provided as a Source Data file.

speciation or higher rates of extinction. Instead, high elevation lineages exhibited high rates of directional migration to lower elevations. This may suggest that the habitats suitable for high mountain plants were not severely reduced during glacial periods but rather shifted downwards. Such shifts instead of loss in habitats may thus have resulted in relatively weaker and more lineage-specific declines in diversification rates. Strictly speaking, our findings are based on lineages that significantly diversified in the European mountain system, and thus do not contradict the paleontological evidence for massive Pleistocene extinctions in the European lowland flora[51–53]. The high rates of endemism and genetic diversity in low elevation refugia at mountain peripheries[7,8,54–56], and the presence of mountain plants in lowland glacial palynological record (*Androsace sp.*, *Gentiana sp.*, *Saxifraga sp.*, but also *Dryas octopetala*, *Polygonum viviparum* or *Saussurea sp.*[57]) however suggest that the above described migration, diversification and survival processes may represent a general model for other extant mountain plant lineages not included in our study, and perhaps also for other components of

European mountain biotas. In fact, it is interesting to note the similarity between the scenario of lineage migrations suggested by our results and the one reported by [58] on Mediterranean mountain birds.

Why did diversification of some lineage slowdown while other remain steady? Given that we have six cases for comparison and we only observed a slowdown in the Primulaceae family (*Primula* and *Androsace*), we can only hypothesize the answer. A scenario of ecological opportunity driving species diversification following mountain uplift[17,27] could provoke a diversification slowdown after saturation of the available ecological space. But such an effect should be observed in the oldest and most diverse lineages of our study clades, rather than in the youngest and relatively least diverse ones such as *Primula* and *Androsace*. The observed diversification slowdown could alternatively be explained by a decreasing rate of allopatric speciation, as the two lineages that slowed down towards the present (*Primula*, *Androsace*) also show the highest rates of sister species allopatry. The prevalence of allopatric speciation can theoretically lead to an intrinsic slow-down of speciation rates, particularly in lineages with low dispersion capacity, once the spatial setting of species' geographic ranges no longer enhances further genetic isolation[59]. Another explanation could be that Pleistocene climatic oscillations promoted migrations between populations in areas that would otherwise remain isolated, thus reconnecting populations and inhibiting allopatric diversification. Simulations of diversification across dynamically fragmented landscapes indeed showed that more connected landscapes or a faster pace of connection-disconnection events can impede species diversification under certain conditions[23,60].

Allopatric speciation has long been regarded as a prevalent mode of speciation in mountain environments[11,15,36,37,61], and the high importance of allopatric speciation demonstrated by our results is generally consistent with this view. An additional interesting finding of our study is the relatively small spatial scale of allopatric speciation, which did not take place between major mountain ranges such as the Alps or Carpathians, but rather within these ranges. It would be interesting to further study the same lineages with nuclear markers, as spatial structure may sometimes emerge due to the usage of chloroplast-based maternal lineage phylogenies. (e.g.[28]). Although the small-scale allopatric speciation appeared to be the most important mode of speciation (37% of speciation events) in sister species of our study lineages, our results further suggest that bedrock-driven divergence has been another important driver (27%) of sister-species speciations. This contrasts with a previous study[11] finding an almost exclusive mode of allopatric speciation and little bedrock divergence in three European alpine Primulaceae genera. These previous results[11] may however reflect a peculiar situation in Primulaceae - a family containing *Androsace* and *Primula* - the two lineages in our dataset for which we found allopatry to be by far the most important mode of speciation.

We estimated that bedrock shifts account for about one fourth (27% in sister species analyses) or one fifth (19% in ClaSSE analyses) of all speciation events across our study clades. Bedrock or edaphic species differentiation is well known from mediterranean floras[62–65], while in European mountain species it has mostly been studied at the intraspecific level[39], or in studies using non-quantitative comparisons to other speciation drivers[66]. To our knowledge, our result thus provides the first quantitative evaluation of the importance of bedrock-driven speciation in European mountain plants. Still, it remains unclear to what extent these speciation events were truly parapatric, that is with reduced gene flow between adjacent populations due to bedrock adaptation, or whether bedrock specialization originated as a consequence of allopatry. Experimental evidence suggests that new

species can indeed arise due to strong divergent selection of calcareous and siliceous ecotypes, even in the presence of gene flow[67]. Our findings have far reaching implications: it shows that contrary to long-standing expectations, many speciation events may not or not only have been due to selectively neutral geographic divergence, but also due to divergent evolution across a complex geological landscape requiring specific physiological adaptations to the divergent proportions of important nutritional elements[68].

Another important finding of our study is the inference of frequent migration events between bedrock types. Intriguingly, we found that these migrations were asymmetrical, with silicate habitats being more often the source of limestone-dwelling lineages than the other way around. This finding seemingly contrasts with previous literature suggesting that plant adaptation to calcareous soils is more restrictive than adaptation to silicate bedrocks[67,69–71]. The higher pleiotropic fitness costs of adaptation to calcareous bedrock may however prevent lineages from switchback to siliceous habitats, and calcareous habitats might thus function as the evolutionary trap[72,73]. Moreover, the stronger migration from silicates to limestones we detected here may also be linked to geographic contingencies: the peripheral lower mountain systems in Europe may be both calcareous and siliceous, while central high elevation parts of European mountains (such as Central Alps, Central Pyrenees or High Tatras) consist almost exclusively of silicate bedrocks[45]. Due to this triangular relationship, the elevation belts may constitute a confounding hidden factor (*sensu*[74]) in the models of evolutionary assembly across bedrock types, where the inferred migrations from high to low elevations and from siliceous to calcareous bedrocks may partly reflect the same process of historical migration out of the central ranges of the mountain systems, possibly due to Pleistocene glaciations. The interaction between bedrock, elevational niche and possibly habitat type may in fact be rather complex, as demonstrated by *Androsace* and *Phyteuma*, which both deviate from the general trend and yielded greater migration rates towards the silicate bedrocks. In *Androsace* this pattern is likely linked to lineages surviving in high elevation silicate rocky habitats, probably even during Pleistocene (discussed below). In *Phyteuma*, silicate specialists are often species inhabiting sub-alpine grasslands in lower silicate mountains with mild slopes, such as Massif Central, Sudetes or eastern Carpathians, where these lineages may also have survived Pleistocene. More work based on an extended population sampling is certainly needed to further disentangle different historical scenarios of evolutionary assembly of alpine floras.

The evolutionary assembly of our study clades across elevation belts was apparently marked by massive directional migration from high to mid-elevations, causing the mid-elevation habitats to host more species despite speciation rates being roughly equivalent in both elevation belts. This directional migration was likely fostered by Pleistocene glaciations causing downward shifts of communities during glacial periods, which then may have left relictual populations at lower elevations during interglacials. Such climate-induced range shifts may have had two-fold consequences on the dynamics of species diversification. On the one hand, it may have prevented species extinctions due to habitat loss during glacial periods, as discussed earlier. On the other hand, intense altitudinal migration may have hampered speciation across elevation belts, as we did not find evidence for speciation along elevational gradients. This finding makes an important distinction with known mechanisms of species diversification in the Andes or other tropical mountains, where speciation across elevation gradients seems to have been a relatively important driver of plant speciation[5,41–44]. More specifically, high and mid-elevation populations in the European mountain system

may have differentiated at intraspecific levels[38,75], but the strong migration from high to lower elevations in response to the severe Pleistocene climate oscillations has likely prevented the emergence of reproductive barriers necessary for speciation events. Important historical migrations along elevation gradients may thus provide an explanation why species diversification has not been as explosive in the European mountain system as in tropical ones[15,29].

The lineage *Androsace* sect. *Aretia* constitutes a notable exception regarding the general scenario of evolutionary assembly highlighted above, probably because this lineage shows strong affinity to high elevation habitats. In comparison to the others, this lineage shows a greater species richness in high elevations, likely due to directional migration from mid- to high elevation habitats. Moreover, according to the ancestral state reconstruction it also contains two sub-lineages that likely evolved exclusively in high elevation habitats. In fact, several species of *Androsace* stretch to the highest limits of vascular plant life in the European mountain system (above 4000 m a.s.l.,[76]), and in Himalaya (6350 m a.s.l.,[77]). Unlike other study lineages, *Androsace* has likely undergone specific adaptations facilitating the repeated entrance into the harsh adaptive zone of high elevation habitats and continued population persistence at high elevations throughout the Pleistocene. A clear manifestation of such adaptation is that *Androsace* repeatedly developed a dense cushion life form, an architecture seemingly well adapted to plant life at high elevations not only in this genus[78] but also in many other angiosperm lineages[79]. Given the inferred long-term affinity of *Androsace* to high alpine environments, it can be reasonably hypothesized that high-elevation sub-lineages of *Androsace* have survived glacial periods in situ in so-called nunatak refugia—i.e. rocky outcrops at high elevations protruding the glaciers[80]— rather than in peripheral refugia, which may have been the prevalent scenario for the majority of the other study lineages. The survival in the limited area of nunataks could also explain Pleistocene diversification slowdown observed in *Androsace*, although the consequences of nunatak survival for species diversification remains unclear. The high elevation sub-lineages of *Androsace* sect. *Aretia* identified in this study thus constitute a perfect system for further tests of the nunatak survival hypothesis at smaller spatial and phylogenetic scales.

In conclusion, our study provides an unprecedented window to the history of diversification of the temperate mountain flora in Europe. It shows that plant diversification within European mountains was a complex evolutionary process, with a strong interplay between altitudinal migration, allopatric speciation and bedrock adaptation of different lineages. Importantly, the onset of Pleistocene climate did not cause a strong diversification slowdown, as was previously expected, but rather stimulated strong migrations of mountain biota across elevation gradients, particularly towards lower altitudes. We hypothesize that these massive altitudinal migration events on the one hand buffered extinctions due to habitat loss during glacial periods, but on the other hand impeded Pleistocene adaptive radiations across the elevation gradient, as is classically known from tropical mountains. We also found speciation events to be mostly driven by geographic divergence but almost as frequently by bedrock shifts. Overall, the absence of obvious adaptive diversification across the elevation gradient and the prevalence of allopatric speciation likely contributed to the lower richness and slower diversification dynamics generally observed in temperate mountains compared to tropical ones.

## Methods

**Clade selection and sampling.** The study lineages were selected as representative cases of plant species diversification in the European mountains by the following

criteria: they contain more than 20 species in total; they contain at least 10 species inhabiting alpine and nival elevational belts in the European Alps (based on ref. [81]); they were recently subject to taxonomic or phylogenetic revision suggesting that Europe is their center of diversity; they are eudicots; and they do not follow derived life strategies as is myco-heterotrophy, parasitism or carnivory. Within the PhyloAlps consortium, which is an extensive network of collaborating institutions (see Supplementary Note), we sampled 212 out of 251 known ingroup species and well-established subspecies from these lineages (that is 84%), specifically 26 out of 29 known species and subspecies for *Androsace* (90%), 45 out of 50 for *Campanula* (90%), 28 out of 35 for *Gentiana* (80%), all 27 for *Phyteuma*, all 24 for *Primula*, and 62 out of 86 for *Saxifraga* (72%). For details on sample counts, see Supplementary Methods 6. The well-established subspecies were treated and referred to as species throughout the study. The majority of ingroup (sub)species were covered by one sample, but in several cases we included two or more samples for control purposes. For all lineages, we followed the most up-to-date taxonomic treatments and took into account phylogenetic studies providing group circumscription, see Supplementary Methods 6 for details. As outgroups for our analyses, we sampled an extensive collection of species from the families Campanulaceae, Primulaceae, Gentianaceae, Saxifragaceae and Grossulariaceae. The latter one was added to Saxifragaceae because of their close phylogenetic relationships[82] and to compensate the lack of fossil calibration points within Saxifragaceae. The majority of samples were collected in natural habitats and immediately desiccated in silicagel, but in some cases we used collections from botanical gardens or herbarium material, for details on sample identities see Supplementary Data 1 and 2.

**DNA extraction and sequencing**. DNA was extracted from collections of leaf tissues gathered during our sampling campaigns, using DNA extraction kits from Macherey-Nagel (Düren, Germany) and Qiagen (Hilden, Germany). The shotgun libraries were prepared and sequenced with methodology depending on the sequencing facility (see Supplementary Data 1 for sequencing facility used for each sample):

For the samples sequenced in Genoscope (Paris, France), the library preparation protocol applied before sequencing was chosen on the basis of the DNA extraction yield. When available, 250 ng of genomic DNA were sonicated using the E210 Covaris instrument (Covaris, Inc., Woburn, MA, USA) and the NEBNext DNA Modules Products (New England Biolabs, Ipswich, MA, USA) were used for end-repair, 3′-adenylation and ligation of NextFlex DNA barcodes (Bioo Scientific Corporation, Austin, Texas, USA). After two consecutive 1x AMPure XP (Beckmann Coulter Genomics, Danvers, Massachusetts, USA) clean ups, the ligated products were amplified by 12 cycles PCR using Kapa Hifi Hotstart NGS library Amplification kit (Kapa Biosystems, Wilmington, MA, USA), followed by 0.6x AMPure XP purification. When the extraction yielded low DNA quantities, 10–50 ng of genomic DNA were sonicated. Fragments were end-repaired, 3′-adenylated and NEXTflex DNA barcoded adapters were added by using NEBNext Ultra II DNA Library prep kit for Illumina (New England Biolabs). After two consecutive 1x AMPure clean ups, the ligated products were PCR-amplified with NEBNext® Ultra II Q5 Master Mix included in the kit, followed by 0.8x AMPure XP purification. All libraries were subjected to size profile analysis conducted by Agilent 2100 Bioanalyzer (Agilent Technologies, Santa Clara, CA, USA) and qPCR quantification (MxPro, Agilent Technologies), then sequenced using 101 base-length read chemistry in a paired-end flow cell on the Illumina HiSeq2000 sequencer (Illumina, San Diego, CA, USA). On average, 5 billion useful paired-end reads were obtained. An Illumina filter was applied to remove the least reliable data from the analysis. The raw data were filtered to remove any clusters with too much intensity corresponding to bases other than the called base. Adapters and primers were removed on the whole read and low-quality nucleotides were trimmed from both ends (while the quality value is lower than 20). Sequences between the second unknown nucleotide (N) and the end of the read were also removed. Reads shorter than 30 nucleotides after trimming were discarded. Finally, the reads and their mates that mapped onto run quality control sequences (PhiX genome) were removed. These trimming steps were achieved using internal software based on the FastX[83].

For the samples sequenced in Fasteris (Geneva, Switzerland) and BGI (Shenzhen, China), custom commercial protocol for preparation was applied. The prepared libraries were sequenced on Illumina HiSeq2000 sequencer (Illumina, San Diego, CA, USA), using 101 base-length read chemistry in a paired-end flow cell.

**Chloroplast genome reconstruction, gene region selection and sequence processing**. The resulting paired-end reads were used to reconstruct the chloroplast genomes using Org.Asm 1.0.3[84], a de novo organelle assembler based on De Bruijn graph implemented in Python 3. We retrieved complete circular plastomes except for samples belonging to Campanulaceae (*Campanula* and *Phyteuma* lineages) where the assembly typically resulted in several discontinuous contigs that however covered the majority of chloroplast coding regions.

We aimed to work with all chloroplast coding regions that were present in our chloroplast genome assembled sequences. To do so, we detected all open reading frames in our circular and fragmented reconstructions and automatically compared them with a curated database of annotated genes from GenBank, following the Org.Annot procedure implemented in Org.Asm[84]. The non-coding regions used in our study were selected on the basis of their universal phylogenetic informativeness[85] or previous use in one of the focal lineages[12–14,19,86–89]. The non-coding regions were

identified and extracted based on their positioning relative to the neighboring coding regions. Coding and non-coding regions of ingroup and outgroup samples were further filtered in order to minimize missing data in our family-level alignment matrices (see below). The filtering resulted in 72 coding regions, among which 40 are shared across all study families, and 17 non-coding regions among which 5 are shared across all the families, see Supplementary Data 3 for a table of regions and their usage for each family. Samples with missing data for some of the regions were kept in specific cases: when they corresponded to the only representative of an ingroup species or to an outgroup species defining the dating node and at the same time had so large portion of missing data that excluding missing regions would severely limit the resolution of the whole family phylogeny. To make sure that inclusion/exclusion had no impact on resulting topology or dating (with exception of dating-node outgroups), we reran phylogenetic reconstruction analyses without these samples and compared the resulting trees with all-samples phylogenies. The samples with missing data in the alignment matrix are indicated as such in the Supplementary Data 2.

For each of the four families, we aligned the coding regions gene-by-gene using MACSE[90] acknowledging the triplet structure of codon alignment. All the positions were quality filtered by Gblocks 0.91[91] (specifying that the data type corresponded to codon alignments), and concatenated together using FasConCat[92]. The non-coding regions were aligned by Mafft 7[93], quality filtered by Gblocks (specifying that the data corresponded to DNA alignments) and concatenated together. This sequence processing pipeline resulted in high-quality sequence matrices for each family ranging between 35471–47102 bp for coding and 2435–5112 bp for non-coding regions, with less than 3% missing data overall.

**Phylogenetic analyses and molecular dating calibrations**. The resulting coding and non-coding alignments for each of the four families were used for inferring dated phylogenies in BEAST 2.6.2[94]. Non-coding alignments and each codon position in coding alignments were modeled with separate averaging-site models as implemented in bModelTest 1.2.0[95]. We used Yule tree prior and lognormal clock with uniform fossil dating priors on at least two nodes of each family. Note that we applied uniform priors between minimum and maximum bounds because the sparse fossil record of our study clades does not allow applying more informative priors—this therefore constitutes a very conservative approach.

In some cases, the fossil record used for dating calibration allowed multiple interpretations of age and positioning in the phylogenetic tree. For consistency with literature, we specifically used the interpretations mirroring previous influential studies on the respective families[14,16,82,86] for the downstream analyses. To test the robustness of this choice, we however reran the phylogenetic analyses with alternative dating calibration where necessary and explored the differences. For a detailed description of used fossil calibrations and their comparison with alternative interpretations, see Supplementary Methods 1 and Supplementary Software 1 and 2.

The BEAST inference for each family was performed in four independent runs, each having 100 M generations and a burn-in of 30%. We evaluated convergence both visually and by checking whether the effective sample size (ESS) was >100 for all the parameters. We used TreeAnnotator to generate maximum credibility trees. The maximum credibility trees and reduced posterior trees were pruned to only include ingroup species and one individual per species, for details on species tree inference see Supplementary Methods 6. We obtained well resolved species-level phylogenies with 87% of nodes receiving >0.95 posterior probability (see Supplementary Software 1 for maximum credibility trees with node supports).

The phylogenies based on chloroplast genomes provide high resolution and accurate dating due to the large number of orthologous genomic regions with variable mutation rates, and the universality of the bioinformatic pipeline allowing to obtain homogeneous phylogenetic information across virtually any angiosperm group[49]. The limitation of our approach is that chloroplast-based phylogenies are only tracking the evolution of maternal lineages, which might be problematic in clades with frequent hybridization, introgression, or incomplete lineage sorting. The shortcomings of plastome-based phylogeny might have impacted, even slightly, the relative branching times and topologies of reconstructed trees of our lineages (see e.g.[13] for *Saxifraga*), and hence some downstream analyses of species diversification. It is important to point out that different downstream analyses we present in our paper are differently prone to biases related to the usage of chloroplast markers: Time-dependent analyses only depend on distribution of branching events through time[96], and are thus least likely to be systematically influenced by the usage of maternal tree topologies. State-dependent analyses are phylogeny-explicit, but their overall results across multiple clades are also relatively unlikely to be biased by the usage of chloroplast markers in a systematic manner. Moreover, the main result of high mobility of lineages across elevational gradient and bedrocks would likely be underestimated rather than overestimated by usage of presumably less mobile maternal lineages. Perhaps the most bias-prone analyses are sister species comparisons, because they directly depend on terminal tree topologies. The absence of polyphyletic species and the low number of identified paraphyletic species (see Supplementary Methods 6 for details) roughly suggest that neither hybridization and introgression nor incomplete lineage sorting are very common in the terminal parts of our phylogenies. More exact prevalence of hybridization, introgression and incomplete lineage sorting in our study clades should however be investigated in future phylogenomic studies using nuclear markers and denser population sampling, especially if results of interest are specific

tree topologies and fine-scale processes, rather than overall diversification patterns and dynamics we present in this study.

**Ecological and geographic characteristics of species**. The information about bedrock, elevational niche and geographical occurrences for each ingroup species was obtained from the following regional floristic literature: Flora Alpina[97], Flora Iberica[98], Flóra Slovenska[99], Wildpflanzen Siebenbürgens[100], and Flora Srbije[101]. We used published ecological and biogeographical studies focused on certain lineages to obtain additional information about genera *Androsace*[102–104], *Phyteuma*[12] and *Saxifraga*[105]. The information about *Campanula* was in part compiled also based on information available in herbarium specimens, local taxonomic literature and from field observations. In several cases, we contacted local taxonomists and made categorization based on provided information. Where possible, the information from floristic literature was compared and supplemented with previously published[11] or publicly accessible[106] sources of point occurrence data.

The calcareous bedrock niche was defined by the regular presence of species on calcareous, dolomitic or ultrabasic bedrocks, the siliceous bedrock niche was defined by regular presence of species on any bedrocks that do not fall in the calcareous category. For instance, in the case of species covered by Flora Alpina[97], species is considered present to a calcareous niche if regular presence is indicated either from limestones (ca) or serpentinites (ser), and to siliceous niche if regular presence is indicated from silicates (si), intermediate substrates (ca/si) or volcanic rocks (bas). For details on the attribution of bedrock niches to individual species, see Supplementary Data 4.

The mid-elevation niche was defined by regular presence of species in habitats below timberline, i.e. up to subalpine elevational zone *sensu* Flora Alpina[97]. High-elevation niche was defined by regular presence above timberline, i.e. in alpine and nival zone *sensu* Flora Alpina[97]. In the cases in which floristic information did not refer to elevational belts, specifically Flora Iberica[98], we marked species presence in mid-elevation niche if the species was inhabiting habitats lower that 200 m below regional timberline and in high elevation niche if the species was inhabiting habitats 200 m above regional timberline. No such treated species was restricted to the range ±200 m around timberline. For details on the attribution of elevational niches to individual species, see Supplementary Data 4.

In order to compile information about species geographical distribution, we defined the smallest operational geographical units across Europe for which we could get credible presence/absence information of every species in our dataset. Specifically, we used operational geographic units in Flora Alpina[97] for the Alps, and mountain regions based on Körner et al.[107] for other European mountains, with subsequent modifications that better reflect structuring of biogeographic information in local floristic literature (see Supplementary Methods 5). We used this approach rather than grid-based processing of point occurrence data (as e.g. in ref. [11]), because of imbalanced point data quality depending on the country and broader European region. For the purpose of ClaSSE models, operational geographic units were merged into five major mountain regions of Europe and surrounding areas (Fig. 1). For details on operational geographic units and how they were merged into five major regions, please refer to Supplementary Methods 5 and Supplementary Data 4.

**Tempo of species diversification**. Based on the inferred phylogenies of each lineage, we fitted five models depicting different temporal dynamics of species diversification (see Table 1). For the models with temperature-dependent speciation or extinction, we used exponential dependence on $^{18}O$ isotope ratio time-series from Greenland ice cores[21]. To accommodate for phylogenetic uncertainty, we fitted every model on 100 trees randomly selected from the Bayesian posterior distribution of phylogenetic trees, and report either the whole distribution (parameter estimates) or median of values (AIC comparisons).

To explore the overall diversification dynamics across all study lineages, we developed a multi-clade framework to fit the above-described diversification models across multiple evolutionary lineages simultaneously. We assumed that each of our lineages is an independent realization of shared diversification dynamics, which allowed us to construct the joint likelihood functions of temporal diversification models as a product of the likelihood functions of each of the six lineages, and harnessing the lineage-specific parameters into shared values. A similar approach with joint likelihood was previously used for state-dependent diversification models[47,108]. We optimized model parameters using the joint likelihood function with a simplex routine, equivalent to the default implementation of single-lineage models in the R package RPANDA 1.5[109] (see Supplementary Methods 2 for implementation details). All calculations using R packages were performed in R 3.5.2.

To test for time- and temperature-dependence of diversification in both multi-clade and single-lineage analyses, we calculated AIC difference (AICdiff) between constant speciation and constant extinction model vs. the other respective models (Table 1). The constant speciation and constant extinction model is nested in the other respective models, and has one less degree of freedom. AICdiff = −2 thus suggests that the focal model parameter does not improve likelihood at all, AICdiff = 0 suggests that both models are equally valuable from information-theoretic point of view, whereas AICdiff>2 suggests substantial support for the focal model[110], i.e. that the focal model would be outperforming the null model

even if there was one completely non-informative parameter added on top of the focal one. We used equivalent interpretation of AICdiff values also in other AIC comparisons throughout the paper.

To test whether the dynamics of diversification are indeed shared across the six lineages as assumed by the multi-clade models, or whether they quantitatively differ among them, we compared the AIC of the multi-clade model with the respective sum of AIC values across the six lineage-specific models. Such a comparison is meaningful, because the sum of AIC values of the six lineage-specific models is equal to the AIC calculated from a model with joint likelihood function, but with each model parameter kept lineage-specific.

The time-dependent diversification models were recently criticized due to identifiability problems[46]. In our analyses we address this issue in several ways: First, we use diversification models corresponding to explicit hypotheses of past diversification dynamics, rather than hypothesis free approaches that were the main subject of criticism. Second, following the recommendations in ref. [46], we use parametrization where only speciation or only extinction is variable in time, and we interpret their results acknowledging that speciation or extinction variability may fall in the same congruence classes. Finally, we perform validation tests and a sensitivity analysis with values realistic for our dataset to show that both single-lineage and multi-clade models correctly identify parameter values from the simulated data (see Supplementary Methods 2 and 3).

**Evolutionary assembly across bedrock types, elevation belts and geographic regions**. We used cladogenetic state-dependent speciation-extinction models[47,48] (ClaSSE) to study separately how diversification and migration occurred across bedrock types, elevation belts and regions. In the models for bedrock types and elevational belts we used parametrization equivalent to GeoSSE[47] that attributes one of 3 states to each species—exclusive for one bedrock type or elevation belt; exclusive for another bedrock type or elevation belt; present in both bedrock types or elevation belts. For the model for geographic regions we developed a generalization of GeoSSE for more than two regions (see Supplementary Methods 4 for script). In this generalization, the number of model states is growing exponentially with number of regions, which prevented us from using a more detailed division of European mountain system than 5 regions (and thus 31 states). All analyses were run with the maximum credibility phylogenies for each of the six focal lineages. Similarly to time-dependent diversification models, we ran all models for each lineage separately, and also a multi-clade model with parameters shared for all the lineages, using the same procedure of likelihood multiplication as described for time-dependent models and equivalent to one in refs. [47] or[108].

For each evolutionary assembly model (on bedrock types, elevation belts, regions), we performed a series of AICdiff comparisons of maximum likelihood model fits to test the importance of different model features: (i) presence state-change speciation (following the terminology of ClaSSE models[48], that is, speciation associated with splits of ancestor species occupying both bedrock or elevation types, or multiple regions), (ii) difference of constant-state speciation rates for each bedrock type, elevation belt or region, (iii) difference of extinction rates for each bedrock type, elevation belt or region, (iv) directionality of migration between bedrock types, elevation belts and regions. The inferred best model was subsequently rerun in a Bayesian setup in order to obtain probability envelopes of parameter estimates, using slice MCMC sampler run for 10,000 iterations and a burn-in of 5000 iterations. The proportions of state-change to all speciation events were obtained by multiplying the present-day numbers of species in different model states (e.g. calcareous specialist, siliceous specialist, bedrock generalist) by their respective speciation rates. The analyses were performed using the R package diversitree 0.9-11[111].

The SSE models are known to suffer from elevated Type I errors when testing constant-state diversification rates differences between states, and various correction strategies were proposed to address this issue[74,112–116]. However, this issue is relevant mostly when the evolutionary states are stable[74,112], which is clearly not the case of our study plant lineages with the important role of migration and state-change speciation across bedrocks, elevational belts and regions. Moreover, all constant-state speciation and extinction rate differences presented in this paper turned unsupported or only weakly supported even without these corrections, and are interpreted as such. Instead, we used the SSE framework for testing and interpreting rates of migration and state-change speciation, and subsequent ancestral state reconstructions, where the SSE methodology is adequate[117]. For estimating state-change dynamics and ancestral state reconstructions, the SSE methodology is known to statistically outperform the alternative approaches not explicitly accounting for diversification dynamics, as is BioGeoBEARS[118] or Mkn model[117]. As recommended[74,115], we also tested the ability of used single lineage and multi-clade SSE to correctly recover parameters, using simulated datasets resembling our data (Supplementary Methods 4).

**Sister-species overlap in geographical and ecological space**. We performed an overlap analysis between sister species in order to address the importance and frequency of allopatric speciation at two different spatial scales, along with speciation between bedrock types and elevation belts. To do this, we identified all sister species pairs in maximum credibility phylogenies. For each of the pairs we calculated the Schoener's D niche overlap index[119] for 87 operational geographic units (fine scale geography), five European mountain regions (coarse scale

geography), elevational belts (high elevation vs. mid-elevation) and bedrocks (calcareous vs. siliceous). The overlap estimates were calculated using the R package spaa 0.2.2[120].

**Reporting summary**. Further information on research design is available in the Nature Research Reporting Summary linked to this article.

## Data availability

Raw genomic data generated in this study are available in European Nucleotide Archive under project accession codes https://www.ebi.ac.uk/ena/browser/view/PRJEB43865, https://www.ebi.ac.uk/ena/browser/view/PRJEB48693, https://www.ebi.ac.uk/ena/browser/view/PRJEB48874 and https://www.ebi.ac.uk/ena/browser/view/PRJEB50489. Accession codes for individual samples are provided in Supplementary Data 1. Species-level phylogenies, both maximum credibility and posterior samples, are available in Supplementary Software 1. Geographic and ecological characteristics of different species are available in Supplementary Data 4. Source data are provided with the paper.

## Code availability

Custom R functions used for fitting multi-clade comparative models are available as a GitHub repository containing an R package (https://github.com/smyckaj/multidiv/tree/6clades, https://doi.org/10.5281/zenodo.6341727 [121]). Examples with simulated data are available in Supplementary Methods 2 and 4.

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

## Acknowledgements

This paper is dedicated to the memory of Prof. Serge Aubert (1966-2015) who devoted his career to the study of alpine plants and was an invaluable source of motivation for the PhyloAlps project. We thank P. Taberlet, F. Condamine, C. Graham, G. Schneeweiss, P. Schönswetter, N. Alvarez, F. Kolář, M. Slovák, T. Herben, D. Storch, M. Elias and L. Pollock for helpful comments, guidance and insights at different stages of the project, M. Smyčková, D. Požárová, M. Kolářová, J. Pilátová, T. Figura, L. Boulangeat, A. Frattaroli, P. Koutecký, C. Lagaye, J. Šibík, P. Anastasiu, L. Filipas, R. Giordano, P. D. Turtureanu, A. Guttová, J. Kučera, M Valachovič, I. Hodalová, J. Kochjarová, E. Štubňová, A. Pleceníková, S. Španiel, G. Casazza, S. Wipf, R. Geremia, P. Choler, N. Passalacqua, L. Hugot, T. Engelskjon, T. Alm, S. Eie, H. Johansen, H. Edvardsen, E. W. Hanssen, C. Bay, B.-G. Osterkloft and L. Sáez for help with sampling, and M. K. F. Merkel and Y. Lammers for help with molecular and bioinformatic analyses. The National park of High Tatra, the DNA Bank of the Natural History Museum Oslo, Tromsø Museum, University in Innsbruck, St Andrews Botanic Garden and the Botanical Garden of the Charles University in Prague kindly provided us plant materials. The research was funded by the joint ANR-SNF project Origin-Alps (ANR-16-CE93-0004, SNF-310030L_170059, S.L. and N.E.Z.) and the OSUG@2020 labex (ANR10 LABX56, S.L.). The sequencing was performed within the framework of the PhyloAlps project, funded by France Génomique (ANR-10-INBS-09-08, P.W.), and the PhyloNorway project funded by the Research Council of Norway (226134/F50, I.G.A.) and the Norwegian Biodiversity Information Centre (14-14, 70184209, I.G.A.). Bioinformatics and statistical analyses were carried out with the GRICAD infrastructure (https://gricad.univ-grenoble-alpes.fr). The sampling campaign and preliminary genomic analyses were partly funded by the European Research Council under the European Community's Seventh Framework Programme FP7/2007-2013 grant agreement 281422 (TEEMBIO, W.T.) and by the SNF grant (31003A_149508, N.E.Z.), the sampling campaigns in the Balkans and in the Iberian Peninsula were funded by the French Ecological Society (SFE²) awarded to J.S., and by the Systematics Research Fund 2016 of the Linnean Society of London and the Systematics Association awarded to C.R. J.S. was supported by a grant from the Doctoral school of Chemistry and Life Sciences within the Univ. Grenoble Alpes, by the ANR project Sphinx (ANR-16-CE02-0011, S.L.) and by Czech Science Foundation (GAČR 20-29554X, J.S.). K.Š. was supported by Charles University project GAUK (815516, K.Š.), the Mobility funds of Charles University and a long-term research project of the Czech Academy of Sciences (RVO 67985939, K.Š.).

## Author contributions

J.S., C.R., and S.L. designed the study, J.S. performed the analyses, and J.S., C.R., and S.L. wrote the first draft of the manuscript. J.S., C.R., M.B., R.D., K.Š., N.E.Z. W.T., I.G.A., S.L. and other members of PhyloAlps consortium contributed to the sample collection and its coordination. C.P., M.R., R.D., and JG.V. coordinated and mounted the reference herbarium collection. M.B., A.A., F.D, P.W., F.B., and E.C. designed and performed the wet lab procedures, sequencing and bioinformatic pipelines. The project as a whole was coordinated by S.L. All authors contributed to the final version of the manuscript.

## Competing interests

The authors declare no competing interests.

## Additional information

## the PhyloAlps consortium

Cristina Roquet [1,4], Martí Boleda[1], Adriana Alberti[5,6], Frédéric Boyer [1], Rolland Douzet[7], Christophe Perrier [7], Maxime Rome[7], Jean-Gabriel Valay[7], France Denoeud [5], Niklaus E. Zimmermann [9], Wilfried Thuiller[1], Patrick Wincker [5], Inger G. Alsos [10], Eric Coissac[1] & Sébastien Lavergne[1]

A full list of members and their affiliations appears in the Supplementary Information.

