## [Peer Review File · Nature Communications]

Tempo and drivers of plant diversification in the European mountain systemREVIEWER COMMENTS

Reviewer #1 (Remarks to the Author):

The authors explored the tempo and drivers of species diversification within the European mountains based on multi-clade comparative phylogenetic methods using a full-plastome phylogenomic dataset from six angiosperm lineages. They demonstrated an asymmetrical migration process between different bedrocks and altitudinal belts. Interestingly, they found that the onset of the Pleistocene climate did not cause a substantial diversification slowdown but rather induced migrations of organisms across elevation gradients. The paper is very well written and the findings are significant. I only have some minor comments.

The phylogenies they reconstructed based on chloroplast genomes provide a very high resolution and quality dating with many orthologous genomic data. They also used a well-designed ClaSSE model to study how diversification and migration occurred across bedrock types and elevation belts. However, the core question and hypothesis wasn't explicitly proposed in the introduction, especially the impact of major ecological gradients (e.g. bedrocks, elevation), which need to be explained in the background.

The topic of this paper is much more concentrated on the glacial effect than more deep evolutionary history, so if it would be more consistent with adding this information in the title. Besides, the authors emphasized on 'tempo and drivers of plant diversification in the European mountain system', however, the tempo of plant diversification did not elaborate much in the main text except the Pleistocene period. Considering the ancient evolution of these selected lineages, please give an essential timeframe for the evolutionary history and pattern of the plant diversification in the European mountains.

Reviewer #2 (Remarks to the Author):

This is a well-prepared and nicely presented paper, using state-of-the-art techniques to address consequential questions in plant speciation and biogeography. It is both a worthy and ambitious piece of research, which sets new lines in the sand for our understanding of drivers of speciation in alpine flora.

While the general motivation for this ambitious paper is clear (i.e. what are the drivers for diversification of mountain/alpine floras), no specific predictions or questions are framed in the Introduction. It is implied (110-124) that the paper will address the effects of several parameters: ecology, bedrock, elevation, glaciation and geographic location (range) upon migration rate, speciation and extinction. The most specific statement is that the paper: "aimed at providing a window into the evolutionary history of European mountain plants by inferring the tempo and drivers of speciation of six study plant lineages". My first impression was that there is a lot going on here, and disentangling the effects of different parameters requires extensive regional as well as phylogenetic replication. However, with six major radiations covered across five major ranges, there is probably enough power to say something useful about all of these parameters.

Estimates of crown-age are largely overlapping (140-142) as are diversification rates (160). On this background, the state-change effect of bedrock type on both speciation rate and migration appears to be more significant, as does geographic separation. I should declare that I am not familiar with some of the analyses used to infer the historical processes, so can only comment on the findings at a general level.

A major limitation of the work, appreciated by the authors, is that phylogenies are based entirely on plastid genomes. These can (and do) cross species boundaries through hybridization, which would impact inferred evolutionary relationships. The authors dismiss this process, saying that current species were (never?) poly- or paraphyletic. This is surprising, given the repeated range and contact changes that have happened during the Pleistocene, evident in hybrid zone work by Hewitt, Taberlet and others. Surely there must be some evidence for PAST hybridization in some of the groups used?

Hybridization is a major driver of speciation in plants, prevalent in saxifrages in particular and also recorded in primulas. Even if all current species are wholly monophyletic, that doesn't mean that past hybridization hasn't occurred. I guess that this would tend to add noise to results, rather than bias, but I do wonder about the effect on inferences about migration. The authors have been a bit too dismissive of this issue.

The reader is left to infer that all five ranges depicted in Fig. 1 were sampled. I can see no reference to this sampling in the main text (only Supplementary), but it could go in the Abstract. It would also be useful to give species count for each clade (although these could be counted from the leaves of the trees) and a total in the Abstract.

115 "HAS occurred"

140 These estimates of crown age all overlap, so statements about their relative ages are not fully merited.

Reviewer #3 (Remarks to the Author):

This paper deals with tempo and drivers of plant diversification in the European mountains. The manuscript is well written in professional, unambiguous language. The figures are explicative, and the topic is of general interest. The manuscript gives a general and interesting overview of the diversification process in the Alps and the results are supported by a large amount of data and analyses. In my opinion the manuscript is worthy of publication.

Major concerns

I have some concerns about logic flow in Introduction. In some parts the line of reasoning goes back resulting in repetitions. For example, the need to collect high-quality data is said in lines 86-89 and in lines 106-108.

Moreover, it is a little bit unclear if the comparison with tropical mountains is an aim of the paper. The relationship between tropical and temperate mountains is just mentioned at the beginning of Introduction but this topic return as a conclusion. If this comparison is an aim of the article, you may spend few more words in Introduction to explain plant diversification processes in tropical mountain. Otherwise, the tempo and drivers of diversification in European mountains is interesting itself.

The study lineages begun to diversify 40My ago, but the discussion is mainly focused on the Pleistocene. It may be useful to say one more word also on previous periods. Otherwise if you focus the work on Pleistocene testing for a slow down during glaciation, this aim should be more clearly said in the text. This last is anyway a very interesting goal.

Minor concerns

Lines 90-94 The relationship between the idea that saturation of ecological opportunities slows down species diversification in the European Alps and the fact that more theories may explain this phenomenon is unclear. You do not investigate which out of the different theories explain better why saturation of ecological space may slowed down species diversification in the European mountain. The logical consequentiality between the two sentences is not very clear.

Line 118. You detect two exceptions (Androsace and Phyteuma) to the main migration pattern from silicate to calcareous habitats that correspond to migration from high to low elevation belt due to the presence of silicate habitat at high elevation and calcareous habitat at mid elevation. Congruently to this expectation in Androsace you detect an opposite migration (from mid to high elevation) and so a higher species richness at high altitude. This is well discussed in the final part of the discussion. However, Phyteuma that have the same migration pattern than Androsace from silicate to calcareous habitat don't have the same migration pattern from mid to high altitude. This may be worth to be shortly discussed.

Your finding that “habitats suitable for high mountain plants were not severely reduced during glacial periods but rather shifted downwards and up again in the interglacials” seems to be in line with that observed in Mediterranean mountain for birds (see Prodon R, Thibault JC, Dejaifve PA (2002) Expansion vs. compression of bird altitudinal ranges on a Mediterranean island. *Ecology* 83(5):1294–1306. [https://doi.org/10.1890/0012-9658\(2002\)083\[1294:EVCOBA\]2.0.CO;2](https://doi.org/10.1890/0012-9658(2002)083[1294:EVCOBA]2.0.CO;2).) This probably worth a little bit discussion because it seems to be a more general pattern.

You find a higher migration rate from silicates to limestones and from high to low altitude. You argued that this may reflect the same process because central high elevation parts of European mountain ranges consist of silicate bedrocks, while peripheral lower mountain systems are mainly calcareous. However, you find both that bedrock shift is an important driver in mountain plant speciation (lines 312-314) but shift between elevation belt isn't. This seems in contradiction with the congruence between bedrock type and altitudinal belt.

At lines 181-181 you shortly explain how you assessed migration. However, the term anagenetic may be a little bit misleading, referring to changes occurring within a lineage. A term clearly referred to the level used to assess migration (within species or taxa, I suppose) may make the text more easily comprehensible. It is crucial to understand how you assessed the migration.

At lines 262-265 you say that previous studies on European mountain detected a diversification slowdown in European mountains while you detected a different pattern. How can you explain this difference?

Lines 272-274. Even if an upwards migration is reasonable, you detected only a downward migration.

At line 347 you spoke about the effect of range shift on diversification dynamics relating to species extinction and speciation. However, this process may also have affected intraspecific differentiation. It could be interesting say a word also on this aspect. Otherwise, you may use a more specific term than diversification dynamics.

Lines 358-360. Why “and”? May the evolution of two sub-lineages at high altitude explain the greater species richness at high elevation?

Line 390. Please could you provide more information about your sampling in this section. How many species have you sampled in each lineage? What is it the percentage of species sampled in each lineage? How many individuals per species? (this is important to understand the reasoning about migration)

In Figure 2A also in *Phyteuma* diversification seems to slow down in the first part of Pleistocene, even if diversification seems to increase at the end of Pleistocene.

Reviewer #1 (Remarks to the Author):

The authors explored the tempo and drivers of species diversification within the European mountains based on multi-clade comparative phylogenetic methods using a full-plastome phylogenomic dataset from six angiosperm lineages. They demonstrated an asymmetrical migration process between different bedrocks and altitudinal belts. Interestingly, they found that the onset of the Pleistocene climate did not cause a substantial diversification slowdown but rather induced migrations of organisms across elevation gradients. The paper is very well written and the findings are significant. I only have some minor comments.

Thank you for these positive comments. We carefully considered all your comments below.

The phylogenies they reconstructed based on chloroplast genomes provide a very high resolution and quality dating with many orthologous genomic data. They also used a well-designed ClaSSE model to study how diversification and migration occurred across bedrock types and elevation belts. However, the core question and hypothesis wasn't explicitly proposed in the introduction, especially the impact of major ecological gradients (e.g. bedrocks, elevation), which need to be explained in the background.

We added more specific questions and expectations about diversification and migration patterns across elevational belts and bedrock types in the third paragraph of Introduction (l 130-136).

The topic of this paper is much more concentrated on the glacial effect than more deep evolutionary history, so if it would be more consistent with adding this information in the title. Besides, the authors emphasized on 'tempo and drivers of plant diversification in the European mountain system', however, the tempo of plant diversification did not elaborate much in the main text except the Pleistocene period. Considering the ancient evolution of these selected lineages, please give an essential timeframe for the evolutionary history and pattern of the plant diversification in the European mountains.

Our study in fact covers the whole period of last 13.7 to 38.1 My, as framed by the oldest crown ages in our dataset, which is why we report it this way. However, the onset of Pleistocene remains the most dramatic climatic event during the study period, and moreover is relatively close to present, which makes our analyses and results more sensitive to this recent era than to the deeper past. As your comment prompts us to better emphasize the timescale of our study, we added a note on deeper climatic dynamics into the second paragraph of the Introduction (l 95-99 and l 111). We also rephrased the the second paragraph of Discussion to better emphasize that our approaches detect temperature-dependent diversification, rather than effect of Pleistocene *per se* (l 285-303). At the current stage, we prefer not to change the title of the study, as we believe that the Pleistocene focus is clear from keywords and the abstract but on the other hand is not central to our study. In case you and the editor have a strong expectation about this, we could change the title but we would rather not artificially reduce the scope of the paper by mentioning Pleistocene in the title.

Reviewer #2 (Remarks to the Author):

This is a well-prepared and nicely presented paper, using state-of-the-art techniques to address consequential questions in plant speciation and biogeography. It is both a worthy and ambitious piece of research, which sets new lines in the sand for our understanding of drivers of speciation in alpine flora.

Thank you for your kind note and your insightful comments below.

While the general motivation for this ambitious paper is clear (i.e. what are the drivers for diversification of mountain/alpine floras), no specific predictions or questions are framed in the Introduction. It is implied (110-124) that the paper will address the effects of several parameters: ecology, bedrock, elevation, glaciation and geographic location (range) upon migration rate, speciation and extinction. The most specific statement is that the paper: “aimed at providing a window into the evolutionary history of European mountain plants by inferring the tempo and drivers of speciation of six study plant lineages”. My first impression was that there is a lot going on here, and disentangling the effects of different parameters requires extensive regional as well as phylogenetic replication. However, with six major radiations covered across five major ranges, there is probably enough power to say something useful about all of these parameters.

We better specified our aims in the second and third paragraph of the Introduction, reflecting also the comments from other reviewers (l 111-115 and l 130-136). In general, we are aware that the picture we tend to draw is complex, with temporal dynamics of species diversification interacting with geographic and niche evolution processes. To make sure that the data we use have enough power to inform us about all these processes, we validated our approaches with simulations presented in the appendices SM2, SM3 and SM4 and referenced in Results and Discussion. We rephrased the respective part of Results (l 203-205) to better emphasize that the informativeness of our dataset with respect to the presented results was tested using simulated data. Moreover, we now also reemphasize this in the Discussion (l 279-281).

Estimates of crown-age are largely overlapping (140-142) as are diversification rates (160). On this background, the state-change effect of bedrock type on both speciation rate and migration appears to be more significant, as does geographic separation. I should declare that I am not familiar with some of the analyses used to infer the historical processes, so can only comment on the findings at a general level.

For time-dependent models, it is indeed true that our results are in line with null expectations, i.e. we do not reject the null hypothesis that diversification rates are independent on past temperatures. We however consider this result extremely interesting and we strictly interpret our results accordingly throughout the manuscript. We also ran a series of sensitivity analyses to confirm that if the alternative time-dependent processes took place, we would likely discover them with our data size and analytical approaches (Supplementary methods SM2 and SM3). For inter-clade comparisons, we did find statistical support for different diversification dynamics between some of the clades (e.g. *Primula* nad marginally *Androsace* decelerate while other four clades not, see Results l 175-188, Table 1 and Figure 2B), even though differences may not be obvious from a mere graphical examination. This approach was also validated through sensitivity analyses. In response to this comment, and also comments from other reviewers, we rephrased the respective part of Discussion (l 285-301) to make these points clear, and to clarify the role of sensitivity analyses in our interpretation.

A major limitation of the work, appreciated by the authors, is that phylogenies are based entirely on plastid genomes. These can (and do) cross species boundaries through hybridization, which would impact inferred evolutionary relationships. The authors dismiss this process, saying that current species were (never?) poly- or paraphyletic. This is surprising, given the repeated range and contact changes that have happened during the Pleistocene, evident in hybrid zone work by Hewitt, Taberlet

and others. Surely there must be some evidence for PAST hybridization in some of the groups used? Hybridization is a major driver of speciation in plants, prevalent in saxifrages in particular and also recorded in primulas. Even if all current species are wholly monophyletic, that doesn't mean that past hybridization hasn't occurred. I guess that this would tend to add noise to results, rather than bias, but I do wonder about the effect on inferences about migration. The authors have been a bit too dismissive of this issue.

We are indeed aware of this limitation and agree that we have been a little too dismissive. It is true that hybridization has been documented in some clades that are the focus of our study and could have impacted some of our results. We now acknowledge more clearly this limitation in the first paragraph of the Discussion (l 269-275). We also modified the Discussion section on spatial scale of allopatric speciation (l 339-341), where we specifically acknowledge that our result might have been influenced by the usage of maternally inherited markers. We have also rewritten the respective Methods section (l 485-509) to be less dismissive: we now cite here evidence of past hybridization in *Saxifragas*, we discuss limitations of chloroplast-based phylogenies with respect to the specific analyses we use, and we also point out the future use of nuclear markers as an interesting study perspective with this regard.

The reader is left to infer that all five ranges depicted in Fig. 1 were sampled. I can see no reference to this sampling in the main text (only Supplementary), but it could go in the Abstract. It would also be useful to give species count for each clade (although these could be counted from the leaves of the trees) and a total in the Abstract.

We added a note in Figure 1 caption, saying that we sampled a 84% of species in study lineages and that we sampled across all five mountain regions. We added the total count of ingroup species in the abstract, and by-clade counts of sampled species in the Methods (l 444-447).

115 "HAS occurred"

We rephrased the whole sentence (l 123-125), thank you.

140 These estimates of crown age all overlap, so statements about their relative ages are not fully merited.

We added a note about the observed overlap between crown ages of different clades, and rephrased the sentence to only point out that *Primula* and *Androsace* were probably younger than the other four lineages (l 153-157).

Reviewer #3 (Remarks to the Author):

This paper deals with tempo and drivers of plant diversification in the European mountains. The manuscript is well written in professional, unambiguous language. The figures are explicative, and the topic is of general interest. The manuscript gives a general and interesting overview of the diversification process in the Alps and the results are supported by a large amount of data and analyses. In my opinion the manuscript is worthy of publication.

Thank you very much for this positive assessment.

Major concerns

I have some concerns about logic flow in Introduction. In some parts the line of reasoning goes back resulting in repetitions. For example, the need to collect high-quality data is said in lines 86-89 and in lines 106-108.

These two statements are indeed a bit redundant, so we removed the second statement which occurred at the end of the second Introduction paragraph. The end of this paragraph now reads as follows (l 112-117): “Documenting the tempo of alpine plant diversification in European mountains therefore requires estimating accurate divergence times for a set of distinct plant clades. This would allow to carefully assess how diversification rates have varied through time and also between regions and environments that have been differently impacted by Pleistocene climate and glacial oscillations.” Note that we modified the Introduction in several places to improve the flow of the text, also with respect to other suggestions made by you and the other reviewers.

Moreover, it is a little bit unclear if the comparison with tropical mountains is an aim of the paper. The relationship between tropical and temperate mountains is just mentioned at the beginning of Introduction but this topic return as a conclusion. If this comparison is an aim of the article, you may spend few more words in Introduction to explain plant diversification processes in tropical mountain. Otherwise, the tempo and drivers of diversification in European mountains is interesting itself.

For a formal comparison of tropical vs temperate plant clades, one should perform simultaneous phylogenetic inference for both. This will hopefully be possible in the future using our dataset along with others. Indeed, our main motivation was to provide an evidence from a temperate mountain system, where the diversification dynamics are generally underexplored. Nevertheless, a verbal comparison of our results with the ones obtained in other alpine mountain systems seems very interesting to us, and we think that our results are in strike contrast to what is already know for certain tropical plant clades. To make clearer where this interpretation comes from, the core Introduction paragraph introducing our study’s incentives now clearly contrast processes that are assumed to be general or temperate-specific (l 123-125 and l 130-136).

The study lineages begun to diversify 40My ago, but the discussion is mainly focused on the Pleistocene. It may be useful to say one more word also on previous periods. Otherwise if you focus the work on Pleistocene testing for a slow down during glaciation, this aim should be more clearly said in the text. This last is anyway a very interesting goal.

We added more information about the pre-Pleistocene dynamics into the Introduction and Discussion (as also suggested by Reviewer #1). Our inference indeed covers the whole period of the last 13.7 to 38.1My, as framed by the oldest crown ages of our dataset. However, the Pleistocene is the most dramatic era in terms of temperature changes, and is relatively close to present. This is why our analyses have more power to detect Pleistocene-related processes than changes that occurred in the deeper past. We hope that our modifications of Introduction (l 95-99 and l 111) and Discussion (l 285-303) rectify the balance a bit more in favor of pre-Pleistocene events.

Minor concerns

Lines 90-94 The relationship between the idea that saturation of ecological opportunities slows

down species diversification in the European Alps and the fact that more theories may explain this phenomenon is unclear. You do not investigate which out of the different theories explain better why saturation of ecological space may slowed down species diversification in the European mountain. The logical consequentiality between the two sentences is not very clear.

The start of this paragraph was indeed not perfectly clear. Thanks for spotting this. As suggested, we have now removed the first sentence mentioning ecological opportunities and saturation of ecological space (l 91-92).

Line 118. You detect two exceptions (Androsace and Phyteuma) to the main migration pattern from silicate to calcareous habitats that correspond to migration from high to low elevation belt due to the presence of silicate habitat at high elevation and calcareous habitat at mid elevation. Congruently to this expectation in Androsace you detect an opposite migration (from mid to high elevation) and so a higher species richness at high altitude. This is well discussed in the final part of the discussion. However, Phyteuma that have the same migration pattern than Androsace from silicate to calcareous habitat don't have the same migration pattern from mid to high altitude. This may be worth to be shortly discussed.

We added a few sentences on these exception into the Discussion paragraph on bedrock related migrations (l 376-384).

Your finding that “habitats suitable for high mountain plants were not severely reduced during glacial periods but rather shifted downwards and up again in the interglacials” seems to be in line with that observed in Mediterranean mountain for birds (see Prodon R, Thibault JC, Dejaifve PA (2002) Expansion vs. compression of bird altitudinal ranges on a Mediterranean island. Ecology 83(5):1294–1306. [https://doi.org/10.1890/0012-\(2002\)083\[1294:EVCOPA\]2.0.CO;2](https://doi.org/10.1890/0012-(2002)083[1294:EVCOPA]2.0.CO;2).) This probably worth a little bit discussion because it seems to be a more general pattern.

We now mention the study as an evidence for similar patterns in Mediterranean mountain birds (l 316-318).

You find a higher migration rate from silicates to limestones and from high to low altitude. You argued that this may reflect the same process because central high elevation parts of European mountain ranges consist of silicate bedrocks, while peripheral lower mountain systems are mainly calcareous. However, you find both that bedrock shift is an important driver in mountain plant speciation (lines 312-314) but shift between elevation belt isn't. This seems in contradiction with the congruence between bedrock type and altitudinal belt.

This is a good point. The relationship between bedrock and elevation is rather triangular than simply positive as we originally claimed – the highest elevation mountain ranges in Europe are almost exclusively silicate, but low elevation ranges can have both silicate and calcareous bedrocks. As a consequence, migration patterns between bedrocks can be partly influenced by elevational dynamics, but at the same time, bedrock driven speciation may have sometimes occurred independently of elevation. We rephrased the respective sentence in Discussion about correlation between bedrock and elevation to point out this triangularity (l 369-373).

At lines 181-181 you shortly explain how you assessed migration. However, the term anagenetic may be a little bit misleading, referring to changes occurring within a lineage. A term clearly referred to the level used to assess migration (within species or taxa, I suppose) may make the text more easily comprehensible. It is crucial to understand how you assessed the migration.

By “anagenetic change” we meant a species-level change not being linked to speciation events and occurring along phylogenetic branches. We discarded the term “anagenetic” and have clarified this in the text, which now reads as follows (l 195-197): “the importance of change of bedrock type, elevation belt or region occurring along branches of the phylogeny, that is without been linked to speciation events (which we term migration)”.

At lines 262-265 you say that previous studies on European mountain detected a diversification slowdown in European mountains while you detected a different pattern. How can you explain this difference?

The cited studies (Boucher et al. 2016 and Kadereit et al. 2004) found a diversification slowdown in *Primula* sect. *Auricula*, *Androsace* sect. *Aretia* and *Gentiana* sect. *Ciminalis*. In *Primula* and *Androsace*, we also found a slowdown, consistently with these studies, but the slowdown was not present in the other four lineages we studied. The slowdown reported by Kadereit et al. 2004 from *Gentiana* sect. *Ciminalis* was possibly overdriven by diversification dynamics in the other two *Gentiana* sections we analyzed together with *Ciminalis*. In any case, these previous studies do not contradict our major conclusion that the Quaternary slowdown of diversification is not universal across European mountain plants. In fact, the conclusion that some lineages slow down while others not, can also be drawn from Kadereit et al. 2004. We rewrote the respective part of Discussion to make this clear (l 285-301).

Lines 272-274. Even if an upwards migration is reasonable, you detected only a downward migration.

Yes, this was misleading, we removed it (l 307).

At line 347 you spoke about the effect of range shift on diversification dynamics relating to species extinction and speciation. However, this process may also have affected intraspecific differentiation. It could be interesting say a word also on this aspect. Otherwise, you may use a more specific term than diversification dynamics.

We replaced “diversification dynamics” with “dynamics of species diversification” to contrast it from “intra-specific differentiation” discussed on lines 390-399.

Lines 358-360. Why “and”? May the evolution of two sub-lineages at high altitude explain the greater species richness at high elevation?

It may, but we agree that the link between these two phenomena is not clear. We rephrased the sentences accordingly (l 403-406).

Line 390. Please could you provide more information about your sampling in this section. How many species have you sampled in each lineage? What is it the percentage of species sampled in each lineage? How many individuals per species? (this is important to understand the reasoning about migration)

Yes, we included it now, also based on the suggestion of reviewer #2 (l 444-447).

In Figure 2A also in *Phyteuma* diversification seems to slow down in the first part of Pleistocene, even if diversification seems to increase at the end of Pleistocene.

It is important to point out that this apparent slowdown takes place after a single event of *Phyteuma* split into its two major groups (*spicatum*-like and *scheucheri*-like). It can thus rather be interpreted as a diversification stasis after the initial split defining the crown age. It is well possible that this slowdown/stasis was related to lowland origin of both these lineages, and the subsequent acceleration around 10 Ma BP was due to parallel adaptation to mountains in both of them. But as the whole stasis pattern is based on a single divergence event at the crown, we consider it too anecdotic to be discussed in the main text.

REVIEWER COMMENTS

Reviewer #1 (Remarks to the Author):

The authors have addressed my comments adequately. I do not have further questions. I recommend this paper to be published.

Reviewer #2 (Remarks to the Author):

The authors have addressed the issues that I raised satisfactorily, but need to proof-read the new text carefully. I saw the errors below:

67 THE Pleistocene
285 HAS caused (singular "onset")
492 LESS likely (?)
498 the low NUMBER of...

Reviewer #3 (Remarks to the Author):

The authors improved the manuscript resolving all weaknesses underlined by reviewers. So, in my opinion the manuscript may be now published.

Reviewer #1 (Remarks to the Author):

The authors have addressed my comments adequately. I do not have further questions. I recommend this paper to be published.

Thank you for your comments, they were really helpful for improving the manuscript.

Reviewer #2 (Remarks to the Author):

The authors have addressed the issues that I raised satisfactorily, but need to proof-read the new text carefully. I saw the errors below:

67 THE Pleistocene

285 HAS caused (singular “onset”)

492 LESS likely (?)

498 the low NUMBER of...

Thank you very much for your positive assesment and careful reading. We fixed these errors and few others we spotted when proof-reading the text.

Reviewer #3 (Remarks to the Author):

The authors improved the manuscript resolving all weaknesses underlined by reviewers. So, in my opinion the manuscript may be now published.

Thank you for your positive assessment and help with improving the manuscript.